# Preparation and Performance Study of Dual-Network Photo-Curable Conductive Silk Fibroin Composite Hydrogel

**DOI:** 10.3390/ma18040779

**Published:** 2025-02-11

**Authors:** Liangduo Li, Xujing Zhang, Yan Xu, Zongheng Shao, Jiahao Ma, Tao Zhu

**Affiliations:** School of Intelligent Manufacturing Modern Industry (School of Mechanical Engineering), Xinjiang University, Urumqi 830046, Chinashixiaoshaoya@163.com (Z.S.); 18599267400@163.com (J.M.); ztao18185@gmail.com (T.Z.)

**Keywords:** dual network, conductive hydrogel, PEDOT:PSS, SF

## Abstract

The printing precision of hydrogels directly determines the mechanical and electrical performance of scaffolds. In this study, poly(3,4-ethylenedioxythiophene)-poly (styrenesulfonate) (PEDOT:PSS) was directly compounded with glycidyl methacrylate-modified silk fibroin (Sil-MA) through a one-pot method to increase the solid content of the printing ink, enhancing its mechanical, electrical, and printability properties. A dual-network photo-curable conductive silk fibroin composite hydrogel (CDMA) was successfully prepared. The results show that the introduction of PEDOT:PSS significantly improved the conductivity of the hydrogel. (The bandgap decreased from 2.36 eV to 1.125 eV, and the maximum conductivity reached 0.534 S/m.) It also enhanced the microscopic 3D network density and mechanical properties of the hydrogel (compressive modulus up to 192 kPa). Moreover, the hydrogel demonstrated good stability during cyclic stability testing, providing a new approach to developing materials capable of high-precision printing with stable electrical performance.

## 1. Introduction

Conductive hydrogels are a new class of composite hydrogels that combine excellent processability, high flexibility, and outstanding electrochemical properties. These materials are widely used in flexible wearable electronic devices and biosensors [1]. Commonly used conductive materials include carbon-based nanomaterials and ionic conductive materials. However, carbon nanomaterials such as carbon nanotubes (CNTs) or graphene nanoplatelets (GNPs) possess a rather high specific surface area, and their high surface energy makes them prone to aggregation, which leads to reduced conductivity [2]. Similarly, in practical applications, ionic conductive hydrogels exhibit uneven ion distribution due to structural changes, resulting in unstable conductivity [3]. Conductive polymers, as a type of polymer material with electrical conductivity between conventional insulating polymers and metals, can be compounded with other materials to address the negative effects of aggregation and dispersibility on conductivity. This allows the fabrication of hydrogels with superior electrical properties [4].

PANI has poor stability and is prone to degradation or loss of conductivity. During oxidation and reduction processes, PPy is susceptible to structural changes, leading to a decrease in conductivity [5]. Among these polymers, PEDOT is an outstanding conductive polymer. Its ethylenedioxythiophene units are alternately arranged with sulfur and oxygen atoms, forming a conjugated system. This conjugated system enables rapid electron transport through π electrons, resulting in high conductivity [6,7].

Furthermore, PSS serves as both a dopant and stabilizer for PEDOT. By doping PEDOT with PSS, hydrophilic groups are introduced to enhance dispersibility. Coulomb interactions assist in constructing a conductive network by dispersing PEDOT in the solution, further improving conductivity [8]. Wang Biao and his team improved the dispersion uniformity of PEDOT:PSS using a constrained phase separation method. This promoted the ordered arrangement of PEDOT chains along cellulose nanofibers, enhancing π–π stacking interactions, forming interconnected PEDOT conductive networks, and achieving a conductivity of 252 S·cm^−1^ [9]. Similarly, Phimchanok Sakunpongpitiporn et al. synthesized highly conductive PEDOT:PSS via chemical oxidative polymerization. They studied the effects of EDOT-to-PSS mass ratios and Na_2_S_2_O_8_ molar ratios on conductivity, finding that the conductivity reached 1879.49 S·m^−1^ when the EDOT:PSS weight ratio was 1:11 and the EDOT:Na_2_S_2_O_8_ molar ratio was 1:2 [10].

Additionally, excellent mechanical properties in hydrogels can also improve PEDOT polymer stacking and enhance the dispersibility of PEDOT:PSS in aqueous phases [11]. Silk fibroin (SF), a polymer with outstanding mechanical properties and a rich three-dimensional network structure, is frequently used to fabricate conductive hydrogels via doping with conductive polymers [12].

Recently, the research group led by Zhan Da prepared composite films by combining silk fibroin with PEDOT:PSS, an organic semiconductor polymer. Their results indicated that this conductive agent preserved the superior electrical performance of the semiconductor without disrupting the structural characteristics of silk fibroin, revealing the effectiveness of doping in producing functional composite materials [13]. To further enhance the printability of conductive hydrogels, Lin Naibo’s team introduced PEDOT:PSS into an Sil-MA solution to prepare photo-curable conductive hydrogels [14]. However, this approach introduced conductive agents directly into the photo-initiator solution, increasing the liquid phase of the printing ink. Consequently, the compressive modulus of the gel was only 27.03 kPa. The decreased mechanical properties further impaired conductivity, limiting the material’s performance.

Therefore, this study aimed to prepare dual-network conductive hydrogels by uniformly dispersing PEDOT:PSS into Sil-MA via a one-pot method. This would reduce the liquid phase in the printing ink, thereby improving both the mechanical and electrical properties of the material. This approach offers a new strategy for designing materials capable of high-precision printing with stable electrical properties.

## 2. Material and Methods

### 2.1. Materials

Silk cocoon sheets (China National Pharmaceutical Group Corp, Beijing, China), anhydrous sodium carbonate (Na_2_CO_3_, purity ≥ 99.5% ), lithium bromide (LiBr, purity: 99.9%), glycidyl methacrylate (GMA, purity: 97%), 3,4-ethylenedioxythiophene (EDOT, purity: 99%), poly(4-styrenesulfonic acid) solution (PSS, purity: 30%), ammonium persulfate (APS, purity: 98.5%), and dialysis bags (MWCO ≥ 13,000 D) were acquired. All experimental reagents and equipment were purchased from Shanghai McLin Biochemical Technology (Shanghai, China).

### 2.2. Methodology

The PEDOT:PSS was synthesized following the procedure reported by Sakunpongpitiporn et al. [10]. First, 5.0 g (4.95 mL) of PSS was slowly added to 100 mL of deionized water and stirred continuously for 1 h. Then, 0.5 g (0.38 mL) of EDOT was slowly introduced into the solution, maintaining an EDOT:PSS weight ratio of 1:11, and stirred for an additional hour. Afterward, 1.67 g of Na_2_S_2_O_8_ was added to the solution, maintaining an EDOT:Na_2_S_2_O_8_ molar ratio of 1:2, and the solution was stirred continuously for 24 h at room temperature (27 ± 1 °C). The color of the solution changed from transparent to dark blue. The mixture was centrifuged to remove impurities, followed by freeze-drying to collect the powder. Finally, the desired solution was prepared via magnetic stirring for 72 h.

Conductive silk fibroin composite hydrogel (CDMA) was developed based on the preparation method of Sil-MA (Figure 1). First, 1 L of 0.05 M Na_2_CO_3_ solution was prepared, and 20 g of silk cocoon sheets were added. The mixture was boiled for 30 min to remove the sericin, followed by washing with deionized water at room temperature to remove excess sericin and sodium bicarbonate. After repeating the degumming process three times, the degummed SF fibers were dried in an oven at 60 °C for 12 h. Next, 10 g of the dried SF fibers were added to 50 mL of a lithium bromide solution (9.5 mol/L), and the mixture was placed in a water bath at 60 °C for 1 h until it fully dissolved into a yellow transparent solution. Then, 3.013 g of glycerol methacrylate was added, and the mixture was stirred for 3 h. The PEDOT:PSS powder was weighed according to the mass ratio of PEDOT:PSS to the degummed silk solution before dissolution. The PEDOT:PSS solution was prepared and then diluted according to the proportions shown in the table. The diluted PEDOT:PSS solution was added to the prepared silk solution, resulting in a photopolymerizable silk fibroin conductive solution with PEDOT:PSS mass ratios of 0.75 wt.%, 0.50 wt.%, and 0.25 wt.%. (The subsequent abbreviations are CDMA-0.75, CDMA-0.5, and CDMA-0.25, respectively.) The solution was placed in dialysis bags with a molecular weight cutoff of 13000 Da and dialyzed with ultrapure water for 4 days to remove the lithium bromide.

Finally, we transferred the dialysis solution into a centrifuge tube and used a high-speed refrigerated centrifuge (TGL16M from KAIDA, Changsha, China) to centrifuge at 9000 r/min for 20 min to remove impurities. Then, we filtered the solution using gauze and performed freeze-drying to obtain three different CDMA powders. The powders were stored in a 4 °C refrigerator for preservation and then used for subsequent experimental investigations.

### 2.3. Fourier Transform Infrared (FT-IR) Spectroscopy

The freeze-dried hydrogels were ground into powder, and test samples were prepared using the KBr pellet method. Fourier transform infrared spectroscopy (FTIR) was conducted at room temperature using a BRUKER VERTEX 70 RAMI spectrometer (Mannheim, Germany). The wavenumber scanning range was set from 400 cm^−1^ to 4000 cm^−1^, with a total of 16 scans performed.

### 2.4. Scanning Electron Microscopy (SEM)

After freeze-drying the samples in a lyophilizer for 48 h, they were cut into small pieces. The cross-sections of the conductive hydrogels were sputter-coated with gold under vacuum conditions using a sputter coater. Subsequently, the cross-sectional morphology of the samples was examined using a Hitachi S-4800 scanning electron microscope (Tokyo, Japan).

### 2.5. Electrical Measurements

Circular hydrogel samples with diameters of 20 mm and thicknesses of 3 mm were fabricated using a photo-curing 3D printing system. The electrical conductivity of the hydrogel samples was measured by employing a four-point probe tester (RTS-9, Guangzhou, China).

### 2.6. Compression Test

Hydrogel samples with a diameter of 10 mm and a height of 10 mm were fabricated by introducing the slurry into silicone molds, followed by ultraviolet (UV) curing. Compression tests were performed at room temperature using a universal mechanical testing machine (model E42.503, Changzhou, China) with a displacement rate of 5 mm/min. Compressive stress–strain curves were obtained to evaluate the mechanical properties of the hydrogels.

### 2.7. Rheology Properties

Cylindrical hydrogel samples with a diameter of 3 cm and a thickness of 2 mm were fabricated. The rheological properties of the hydrogels were assessed at 25 °C using a rheometer (Anton Paar, Austria, model MCR 302). To prevent drying during testing, the edges and surfaces of the hydrogel samples were coated with oil.

### 2.8. Hydrogel Stability Test

One end of the conductive hydrogel was adhered to the electrode, while the other end was connected to conductive tape, and the assembly was used to construct a flexible strain sensor. The resistance changes under pressure were recorded using an electrochemical device, and the following formula was used to calculate the resistance change:$$\Delta R/R_{0}=\left(R-R_{0}\right)/R$$
where *R* represents the resistance of the hydrogel under a specific external force and $R_{0}$ represents the initial resistance value before any pressure is applied.

Next, NaCl (0.8 g), KCl (0.2 g), Na_2_HPO_4_·12H_2_O (3.63 g), and KH_2_PO_4_ (0.24 g) were dissolved in 900 mL of ultrapure water, and the pH was adjusted to 7.4 using concentrated hydrochloric acid. The solution was then diluted to a final volume of 1 L and stored in a 4 °C refrigerator for later use.

In a plastic Petri dish 90 mm in diameter, 50 mL of 0.1 mol/L PBS solution was added. Circular hydrogel samples with diameters of 20 mm and 3 mm were completely immersed in the PBS solution. The Petri dish was covered and placed in a 37 °C constant-temperature incubator. The electrical resistivity of the gel samples was tested continuously (the gels were removed and dried using filter paper before measurement). Each sample was tested three times, and the average values were recorded.

## 3. Results and Discussion

### 3.1. Compatibility Analysis

To assess the feasibility of polymer blending, the repeating units of PEDOT and PSS, along with the Sil-MA molecular model, were first constructed using Materials Studio software (2022, BIOVIA, San Diego, CA, USA). After geometric optimization using the Compass III module in the Forcite package, the compatibility between the materials was evaluated through the Blends calculation. The blending simulation was conducted using the Dreiding force field, and QEq was used to optimize the forces and charges on the models to ensure calculation accuracy. Additionally, both electrostatic interactions and van der Waals interactions were controlled by atom-based interactions. The binding energies of the blends were calculated.

The binding energies between Sil-MA, PEDOT, and PSS were analyzed by comparing the base–base (Ebb), screen–screen (Ess), and base–screen (Ebs) interactions. Figure 2 shows that the peaks of the curves for the Sil-MA/PEDOT and Sil-MA/PSS combinations reached 5903.17 and 5014.61, respectively, within the energy range of −5–0 kcal/mol, indicating a strong bonding strength and higher stability between the materials within this energy range. In Figure 2a, fluctuations in the range of −10–0 kcal/mol may suggest the presence of some unstable interactions within the system, which could affect the stability and miscibility of the blend. In Figure 2b, the peaks of the Ebb and Ebs values for the Sil-MA/PSS combination were more concentrated with a narrower distribution range, indicating more stable interactions between the components. Despite minor differences, the distributions of the three energy curves remained consistent, suggesting that the three materials can achieve effective blending [15].

### 3.2. Electronic Properties Analysis

Further investigation into the electronic properties of the materials was conducted using DMol3 while employing density functional theory (GGA-PBE) calculations combined with first-principles analysis. The calculations, based on geometric optimization and parameter settings, provided a reliable foundation for analyzing the system’s band structure. The obtained band structure data were used to examine the material’s conductivity, bandgap size, and electronic transition characteristics, providing theoretical support for a deeper understanding of the system’s electronic properties and reaction mechanisms.

Three gradient doping systems of Sil-MA with 0.25%, 0.5%, and 0.75% PEDOT:PSS, along with a blank system, were set up for band structure and density of states (DOS) analysis to evaluate their conductive performance. The geometric optimization parameters were set to medium quality, with the convergence criteria including an energy tolerance of $2.0\times 10^{5}$ Ha, a maximum force of 0.004 Ha/Å, a maximum displacement of 0.005 Å, and a maximum step length of 0.3 Å. The GGA-PBE functional was used with the DNP basis set, full electron treatment, and a self-consistent field (SCF) convergence tolerance of $1.0\times 10^{-5}$, with a maximum of 100 iterations. A 3 × 3 × 1 k-point mesh centered at the gamma point was applied.

The results showed that after doping with PEDOT:PSS, the bandgap of Sil-MA decreased from 2.360 eV to 1.125 eV (CDMA-0.25), reaching a minimum of 0.484 eV (CDMA-0.5). The reduction in the bandgap indicates that the material is more prone to electronic transitions, which is beneficial for optimizing the conductive hydrogel’s performance. This is because the π-conjugated system of PEDOT interacts with the electrons of silk fibroin, forming more conductive pathways near the Fermi level and thereby enhancing the conductivity of the gel. Meanwhile, PSS, as a dopant, stabilizes the charge carriers and enhances carrier mobility, significantly improving the gel’s conductivity.

In the subsequent analysis (Figure 3), Sil-MA exhibited many high-energy electronic states in the range of −10–0 eV, with several peaks in the density of states curve. This suggests that electron transitions between energy states may not be smooth, indicating poor conductivity for the material [16]. As the conductive factor content increased, the material’s conductivity improved. Notably, the CDMA containing 0.5% conductive factor exhibited a high density of electronic states (DOS) near the Fermi level. Additionally, the smoother DOS curve in this material indicates that more electronic states are available for carriers to occupy at specific energy levels. The reduced energy requirement for electron excitation from the valence band to the conduction band indicates that CDMA possesses good conductivity [17].

### 3.3. Fourier Transform Infrared Spectroscopy (FTIR) Analysis

FTIR spectroscopy was used to analyze the microstructures of SF, Sil-MA, and CDMA. Significant fluctuations in the spectrum were observed upon the addition of PEDOT:PSS (Figure 4a). The main characteristic absorption peaks of SF appeared at 1637 cm^−1^, 1513 cm^−1^, and 1233 cm^−1^, corresponding to the β-sheet structure or the silk II anti-parallel β-sheet structure’s amide I (C=O), amide II (N-H bending), and amide III (C-N stretching) [18].

The characteristic peak of Sil-MA was primarily observed at 1238 cm^−1^, indicating the formation of a CHOH group due to the ring-opening reaction of GMA. Other minor changes detected in Sil-MA were at 951 cm^−1^ and 1165 cm^−1^, primarily due to the stretching of the CH2 group in glycerol methacrylate. As the amount of GMA increased, these peaks gradually became more pronounced. However, due to the smaller molecular weight of GMA compared with SF, the GMA functional groups were only slightly detected [19].

The characteristic peaks of PEDOT:PSS appeared at 3420 cm^−1^, 1521 cm^−1^, 1437 cm^−1^, and 1375 cm^−1^ [20], corresponding to the O-H stretching vibrations in PSS, C=C stretching vibrations in PEDOT and PSS, symmetric vibrations of the C=C bonds in the five-membered ring of PEDOT, and stretching and deformation of C-C single bonds, respectively. The FTIR spectrum of the PEDOT:PSS/Sil-MA hydrogel showed peaks at 3279 cm^−1^, 1622 cm^−1^, 1514 cm^−1^, 1125 cm^−1^, and 1063 cm^−1^, which were attributed to N-H and C=O stretching vibrations, N-H bending vibrations, and symmetric and asymmetric stretching vibrations of S=O due to hydrogen bonding interactions, respectively. The conjugated C=C vibration peak of the quinone-type structure in the PEDOT:PSS molecular chain exhibited a strong absorption peak, while the conjugated C=C bond of quinones was weaker [21]. This is because the PEDOT:PSS molecular chain was influenced by the protein molecules, causing the resonance structure to change. As the mass fraction of the conductive factor increased from 0.25% to 0.75%, more PEDOT molecular chains underwent a transition from the quinone-type resonance structure to the quinone form. Quinone compounds possess stronger electron-withdrawing properties, and the carbonyl groups in their structure lead to stronger π-π stacking or other intermolecular interactions, which result in the agglomeration of the material [22].

Further secondary structure analysis of the hydrogel revealed that the silk fibroin (SF) is composed of subunits, including the heavy chain, light chain, and glycoprotein p25, which are linked by disulfide and hydrophobic bonds [23,24]. The secondary structure mainly consists of several forms, including the β-sheet, random coil, α-helix, and β-turn forms. By controlling the composition of these secondary structures and their relative proportions, biomaterials with varying morphologies and properties can be produced [25,26]. Among them, the β-sheet structure of SF links SF molecules together to form a 3D network, ensuring the strength of the formed hydrogel [27].

Fitting analysis of its characteristic peaks using Peak software (41, Thermo Fisher Scientific, Waltham, MA, USA) (see Figure 4b–e) revealed that with the introduction of GMA, the number of hydrogen bonds increased, interacting with the random coil in SF, which caused the random coil to arrange into sheet-like structures, forming a more stable β-sheet structure. When PEDOT:PSS was further introduced, the number of hydrogen bonds continued to increase, resulting in the β-sheet content of the gel reaching 44% (Figure 4b). The β-sheet structures can stack on top of each other, leading to a more densely packed 3D network structure [28]. This tight packing reduces the internal voids of the gel structure, enabling it to store and dissipate more energy. On the other hand, the abundance of β-sheet structures promotes the formation of tight hydrogen bonds between molecules, further inhibiting chain breakage and sliding, which enhances the gel’s resistance when subjected to stretching, compression, or shear forces [29].

### 3.4. Macro- and Micro-Structural Morphology Analysis of the Hydrogel

Scanning electron microscopy (SEM) analysis revealed that the pore size of the hydrogel was inversely correlated with the content of the conductive factor; that is, as the content of the conductive factor increased, the pore size of the hydrogel’s 3D network became larger (see Figure 5a). When the PEDOT:PSS content was 0.25 wt.%, the gel exhibited relatively large pores, but no significant conductive network was formed. At a PEDOT:PSS content of 0.50 wt.%, the pore size increased further. This is because PEDOT:PSS can easily mix with the hydrogel and incorporate into its structure, leading to increased distances between crosslinking points and larger pore sizes, thereby forming a distinct double-network structure. When the PEDOT:PSS content reached 0.75 wt.%, PEDOT:PSS agglomerated into block-like structures, reducing the number of through-pores [30].

Further analysis using ImageJ software (1.54, Wayne Rasband, Bethesda, MD, USA) to fit and explore the effects of PEDOT:PSS contents of 0.25 wt.%, 0.50 wt.%, and 0.75 wt.% on the conductive network content showed that the conductive networks amounts were 4.01%, 12.34%, and 59.58%, respectively. This is because larger pore networks are more difficult to stabilize due to the interactions between the fillers, which leads to uneven distribution of the fillers within a network [31]. Consequently, the conductive particles tend to aggregate, causing a significant increase in the conductive network content.

### 3.5. Conductivity Performance Analysis

The conductivity of the conductive hydrogel was characterized using the four-probe method. It was found that the hydrogel achieved the highest conductivity of 0.534 S/m when the content of the conductive factor was 0.50 wt.% (see Figure 5c). This is because as the gel’s light absorbance increased, it was able to capture more photons for reactions, enhancing its mechanical properties. As the mechanical properties improved, the stacking of PEDOT:PSS polymers was optimized, and the dispersion of PEDOT:PSS in the aqueous phase increased, which in turn improved the material’s conductivity [10]. However, when the PEDOT content exceeded 0.50 wt.%, the conductivity decreased significantly to 0.426 S/m.

This result can be attributed to the fact that the conductivity of the hydrogel relies on Sil-MA as a matrix to ensure the uniform distribution of conductive pathways. Beyond this ratio, the aggregation of PEDOT:PSS began to occur, hindering the formation of a uniform conductive network [32]. This aggregation not only affected the overall connectivity of the conductive network but also obstructed the effective transmission of charge, leading to a decrease in conductivity. Further observation of the brightness of a small light bulb can directly reflect the conductivity of the gel. When CDMA-0.5 was connected to the circuit, the light bulb emitted a dazzling glow. However, when CDMA-0.75 was connected to the circuit, the brightness of the bulb decreased. Therefore, to ensure the reliability of the experiment and the optimal performance of the conductive properties, the 0.50 wt.% PEDOT:PSS hydrogel was chosen as the main research subject in subsequent studies to further explore the characteristics of the double-network conductive hydrogel.

### 3.6. Mechanical Properties Analysis of the Hydrogel

In the mechanical testing, CDMA-0.5 powder and Sil-MA powder were each mixed with a photoinitiator to prepare three sets of gel printing slurries with gel mass fractions of 15%, 20%, and 25%. The compression stress–strain curves for the CDMA hydrogel and Sil-MA hydrogel are shown in Figure 6a. The study found that as the gel content increased, the material’s compressive modulus increased from 94 kPa to 161 kPa. However, the compressive modulus of 161 kPa corresponded to a gel content of 20 wt.%. This is because when the gel content in the slurry reached 25 wt.%, the amino acids in the Sil-MA interacted with the excess polystyrene sulfonate (PSS) ions in the PEDOT:PSS through hydrogen bonds or electrostatic interactions. This increased the viscosity of the material and caused air bubbles to form easily during gel printing, leading to a decline in mechanical performance.

However, with the addition of the conductive factor, the material’s light absorbance increased, allowing it to capture more photons, which enhanced the curing of free radicals [33]. As a result, the compressive modulus of the gel with 20 wt.% CDMA-0.5 increased to 192 kPa. Therefore, the optimal ratio of gel to photoinitiator was determined to be 1:4 for subsequent studies.

### 3.7. Rheological Analysis

The viscosity of the hydrogel is a key factor in determining whether the ink can be successfully and stably printed into the desired model. The precursor solution of the ink must maintain a low apparent viscosity to ensure quick and uniform addition to the reservoir. To investigate the impact of the modified gel on the rheological properties of the photopolymerizable composite ink, the storage modulus (G′) and loss modulus (G″) of SF, Sil-MA, and CDMA-0.5 were tested within the strain range of 0.01–1% under a constant angular frequency (10 rad/s). The storage modulus (G′) represents the material’s ability to store elastic deformation, reflecting the material’s stiffness, while the loss modulus (G″) indicates energy dissipation during the viscous deformation process, reflecting the material’s viscosity.

In Figure 6b,c, it can be observed that the storage modulus (G′) and loss modulus (G″) of the SF hydrogel, Sil-MA hydrogel, and CDMA-0.5 hydrogel remained relatively stable, with the G′ values being significantly greater than the G″ values for all three materials. This indicates that the materials could quickly recover to their original shape after stress and did not exhibit noticeable plastic or rheological characteristics due to internal dissipation (i.e., loss modulus) [34]. Among them, the storage modulus of CDMA-0.5 was significantly higher than those of the other two materials, consistently ranging between 2500 and 3000 Pa, indicating that CDMA-0.5 has a strong energy storage capacity with a tightly packed elastic network which can resist deformation caused by external forces. On the other hand, the G′ of SF ranged from 0 to 200 Pa, indicating that SF had the weakest elastic network and was almost unable to store energy. Additionally, the G′ and G″ values of the CDMA-0.5 and Sil-MA hydrogels were higher than those of the SF hydrogels, indicating that these materials can better store energy and reduce energy dissipation. This suggests that the addition of PEDOT:PSS and GMA enhanced the crosslinking network density of the hydrogel.

**Figure 6 materials-18-00779-f006:**
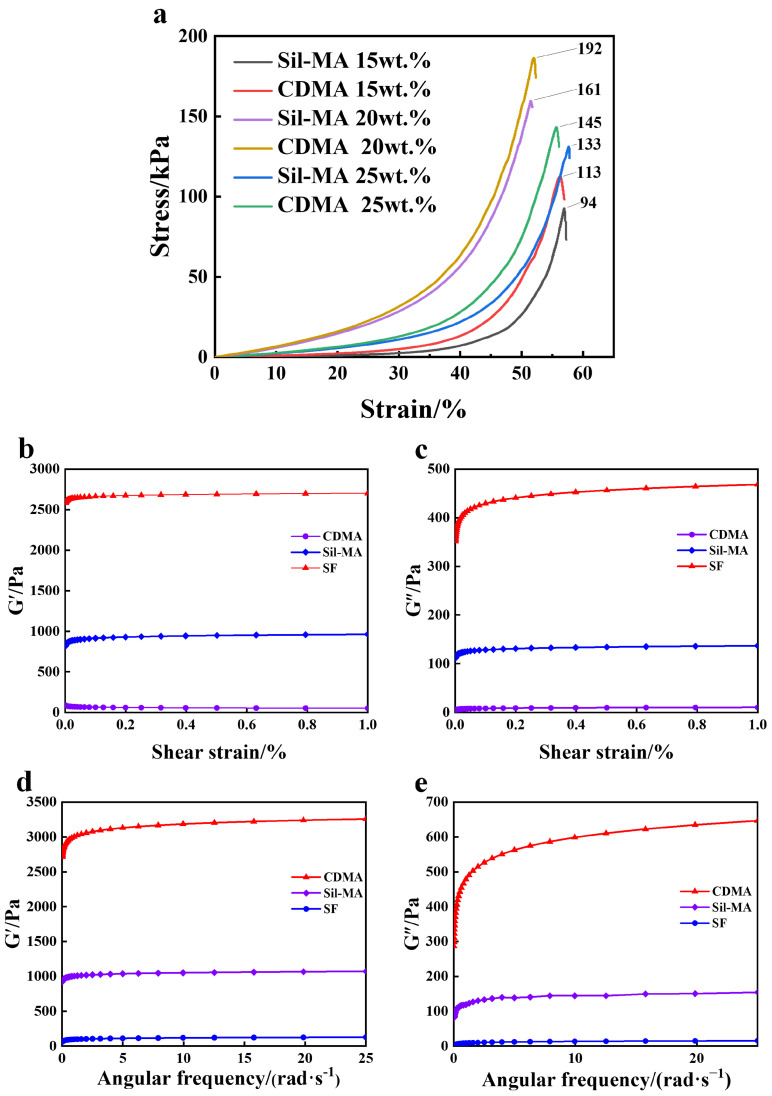
(**a**) Compression stress−strain curves of Sil−MA and CDMA with different photoinitiator ratios. (**b**) G′ of Sil−MA and CDMA in the strain range of 0.01–1%. (**c**) G″ of Sil−MA and CDMA in the strain range of 0.01–1%. (**d**) G′ of Sil−MA and CDMA in the frequency range of 0.158–25.1 Hz. (**e**) G″ of Sil, MA and CDMA in the frequency range of 0.158–25.1 Hz.

Under constant strain (1%) conditions (Figure 6d,e), the G′ and G″ values of the SF, Sil-MA, and CDMA-0.5 hydrogels within the frequency range of 0.158–25.1 rad/s showed minimal stress changes at low frequencies, while shear thinning occurred at high frequencies. This implies the presence of strong crosslinking interactions (such as covalent and hydrogen bond crosslinking) within the networks, which effectively enhances the structural stability of the hydrogel and reduces rheological changes at high shear rates [35]. Throughout the entire frequency range, G′ remained the highest (close to 3000 Pa), with the elastic network of CDMA-0.5 being insensitive to frequency changes, indicating that it can quickly respond to external forces and exhibit significant solid elastic behavior. However, at low frequencies (<5 rad/s), G′ increased slightly before stabilizing, indicating that the network structure became saturated. This suggests that CDMA-0.5 is a material with high viscoelastic properties, making it suitable for applications which require high energy storage and dissipation capabilities.

### 3.8. Conductivity Stability Analysis of the Hydrogel

Figure 7a shows the relative resistance change (ΔR/$R_{0}$) of the hydrogel under different compressive strains. It can be clearly observed that the relative resistance change exhibited a positive correlation with the compressive strain. This is because under compressive strain, the internal structure of the hydrogel deforms, leading to a reduction in conductive pathways and, consequently, a decrease in resistance. The relative resistance increased as the strain increased. When the external force was removed, the hydrogel’s structure returned to its initial state, and the conductive pathways were restored, causing the resistance to return to its original value. The resistance varied within the range of 0–9.14%, maintaining consistency without signal drift and demonstrating stable sensing performance and excellent durability.

Subsequently, the hydrogel was immersed in a 0.1 mol/L PBS solution, and the gel was periodically removed and dried with filter paper to test its conductivity. In this case, $R_{0}$ represents the initial resistance of the thin layer, and *R* is the resistance of the thin layer after the test. From Figure 7b, it can be observed that the resistance of the disc-shaped hydrogel sample remained almost unchanged with the degradation time. However, the quinone structure in the PEDOT exhibited certain photosensitive properties. Light exposure exciteed electrons from the π-π structure to higher energy levels, making the chemical bonds in the molecules unstable and disrupting the original quinone structure. This process also led to reorganization of the conjugated system, ultimately forming a benzene ring structure, which resulted in an increase in the material’s conductivity.

## 4. Conclusions

In this study, PEDOT:PSS was introduced during the preparation process to fabricate a photopolymerizable conductive hydrogel, which reduced the liquid-phase content of the printing slurry and improved the material’s printability. The results showed that an appropriate amount of PEDOT significantly enhanced both the conductivity and mechanical strength of the hydrogel, particularly when the PEDOT content was 0.50 wt.%. The introduction of PEDOT:PSS facilitated the formation of a dual 3D network structure within the hydrogel, which improved its conductivity and mechanical properties, resulting in the highest conductivity (0.534 S/m) and compressive modulus (192 kPa). Additionally, during the degradation process, the resistance of the CDMA was found to remain relatively stable. Overall, this study provides a novel approach for developing high-precision conductive hydrogel materials. 

## Figures and Tables

**Figure 1 materials-18-00779-f001:**
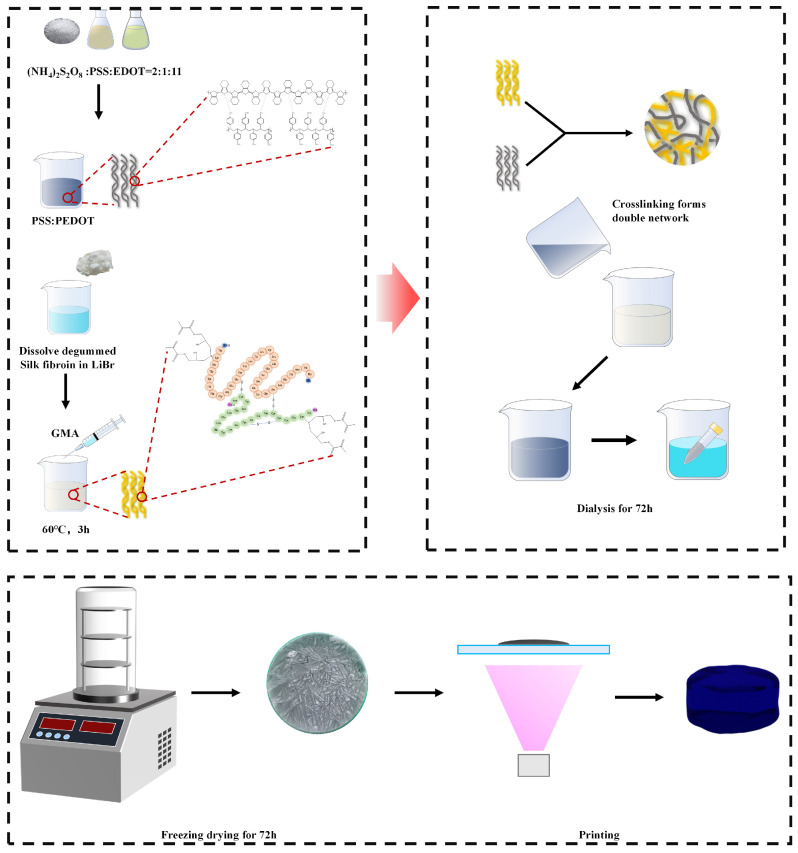
Fabrication process of photo-curable conductive hydrogels.

**Figure 2 materials-18-00779-f002:**
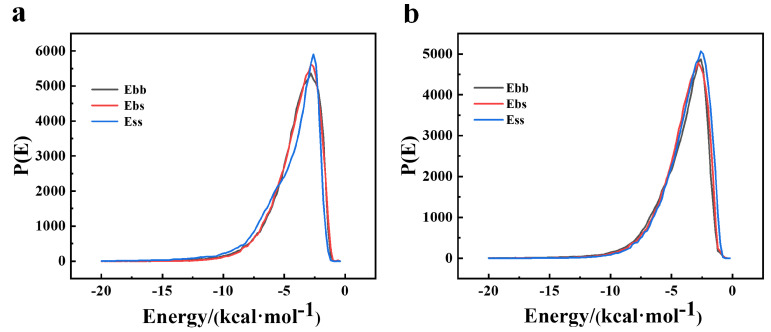
(**a**) Binding energy between Sil−MA and Pedot. (**b**) Binding energy between Sil-MA and PSS.

**Figure 3 materials-18-00779-f003:**
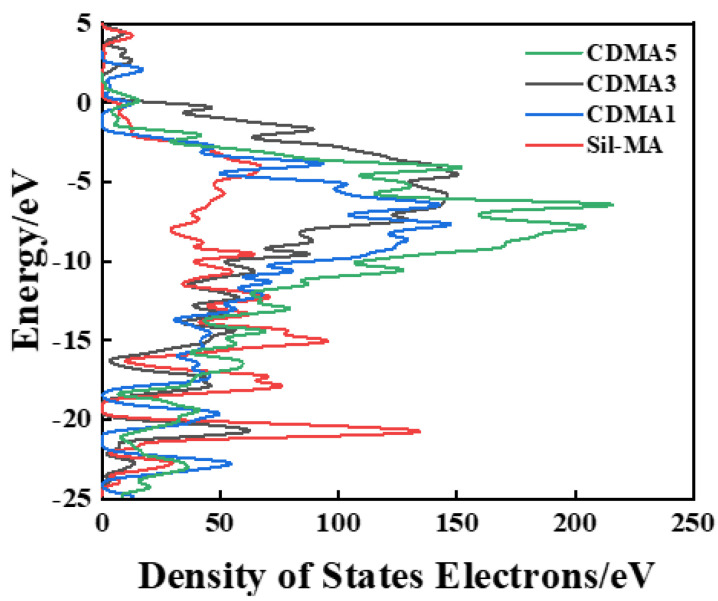
Gel state density of states (DOS) with varying amounts of conductive factors.

**Figure 4 materials-18-00779-f004:**
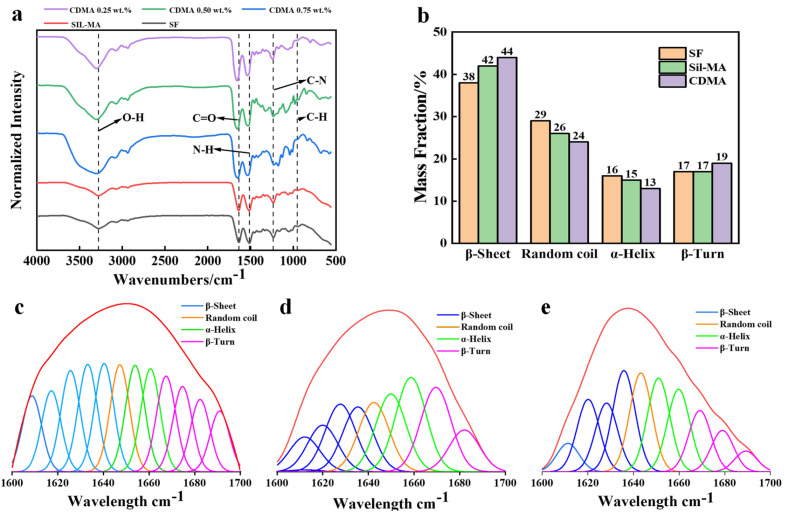
(**a**) FT−IR spectra of the gel before and after modification and doping. (**b**) Peak fitting results and statistical analysis after FT−IR spectrum deconvolution. (**c**) FT−IR spectrum deconvolution and peak fitting results for CDMA (1580–1720 cm^−1^). (**d**) FT−IR spectrum deconvolution and peak fitting results for Sil-MA (1580–1720 cm^−1^). (**e**) FT−IR spectrum deconvolution and peak fitting results for SF (1580–1720 cm^−1^).

**Figure 5 materials-18-00779-f005:**
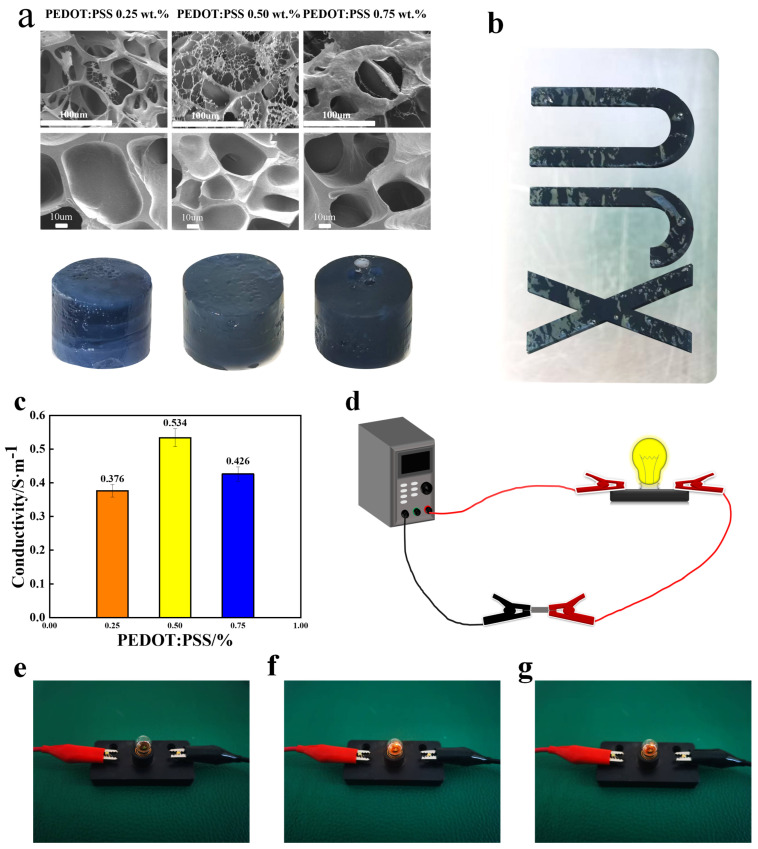
(**a**) Macro- and microstructural images of the gel with different conductive factor contents. (**b**) Printing effect of CDMA-0.5. (**c**) Relationship between the conductivity of the hydrogel and the PEDOT:PSS content. (**d**) Circuit diagram. (**e**) Brightness of the light bulb when CDMA-0.25 was connected. (**f**) Brightness of the light bulb when CDMA-0.50 was connected. (**g**) Brightness of the light bulb when CDMA-0.75 was connected.

**Figure 7 materials-18-00779-f007:**
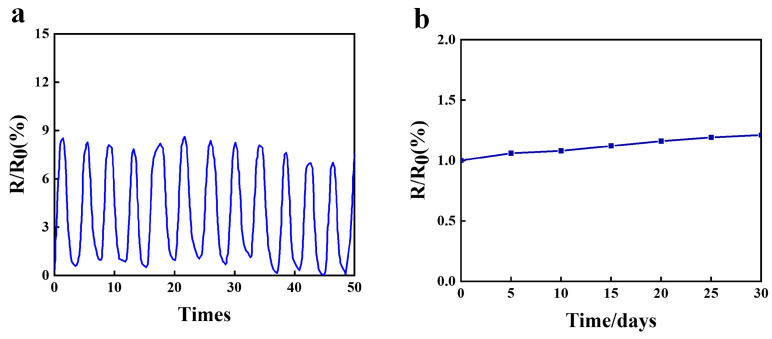
(**a**) Cycling stability test of the conductive hydrogel. (**b**) Relationship between the degradation time and resistance change of the conductive hydrogel.

## Data Availability

The datasets generated and analyzed during the current study are available from the corresponding author upon reasonable request.

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
