# Peer review of "Preparation and Performance Study of Dual-Network Photo-Curable Conductive Silk Fibroin Composite Hydrogel"

_materials, 2025, doi:10.3390/ma18040779_

Round 1

Reviewer 1 Report

Comments and Suggestions for Authors

The manuscript by Li et al. aimed to prepare dual-network conductive hydrogels by uniformly dispersing PEDOT:PSS into glycidyl methacrylate-modified silk fibroin via one-pot method. The theme of this paper is quite original, the results are supported by the experiments very well. Conclusions are clear, and according to the results. This paper brings some important information, which could be useful in the printing industry.

 In my opinion, this manuscript should be published after minor revision:

Abbreviations PEDOT, PSS, CDMA, EDOT, Sil-MA, GMA, PBS should be previously introduced in the main text of the Manuscript, not just in the Abstract.

General, correct examples: “…Na2S2O8…”, “…Na2CO3…”, “…Na2HPO4·12H2O (3.63 g), and KH2PO4 should be replaced with “…Na2S2O8…”, “…Na2CO3…”,  “…Na2HPO4·12H2O (3.63 g), and KH2PO4…”  i.e. includes a subscript.

In section 2. Materials and methods, the authors didn’t provide any information about centrifuge. Please add the product name, model, manufacturer, country, and city.

Figure 1 title should be corrected to “Schemes follow the same formatting.”.

I suggest that the title of Section 2.7. should be replaced with “Rheology properties”.

Figure 3a should be replaced with only Figure 3., in the text and in Figure letter a delete.

In the Section 3.3. line 237 and line 242 “(see Figure 5b, c, d, e) “ and “(Figure 7b)” should be replaced with “(see Figure 4b, c, d, e) “ and “(Figure 4b)”, the discussion refers to Figure 4 and not to Figures 5 and 7.

The authors should correct the title of Figure 5. Instead of CMDA replace CDMA.

Figure 7 should be placed within Section 3.8.

Line 360. The last chapter should be renamed in Conclusion, it is not Materials and Methods.

Author Response

Abbreviations PEDOT, PSS, CDMA, EDOT, Sil-MA, GMA, PBS should be previously introduced in the main text of the Manuscript, not just in the Abstract.

Thank you for pointing this out. I agree with this comment. Therefore, I have removed the abbreviations from the abstract and provided the corresponding explanations in the introduction. This change can be found in the revised manuscript in line 43 on page 2 and in line 83 on page 3.

General, correct examples: “…Na2S2O8…”, “…Na2CO3…”, “…Na2HPO4·12H2O (3.63 g), and KH2PO4 should be replaced with “…Na2S2O8…”, “…Na2CO3…”,  “…Na2HPO4·12H2O (3.63 g), and KH2PO4…”  i.e. includes a subscript.

Thank you for pointing this out. I agree with this comment. Therefore, I have changed them to the corresponding subscripts. This change can be found in the revised manuscript from line 73 on page 2 to line 82 on page 3, and line 138 on page 4.

In section 2. Materials and methods, the authors didn’t provide any information about centrifuge. Please add the product name, model, manufacturer, country, and city.

Thank you for pointing this out. I agree with this comment. Therefore, I have added the corresponding information in line 99 on page 3.

I suggest that the title of Section 2.7. should be replaced with “Rheology properties”

Thank you for pointing this out. I agree with this comment. Therefore, I have updated the title for this section, which can be found in line 126 on page 4.

Figure 3a should be replaced with only Figure 3., in the text and in Figure letter a delete.

Thank you for pointing this out. I agree with this comment. Therefore, I have removed the letter 'a'. This change can be found in the revised manuscript in Line 202 on page 6.

In the Section 3.3. line 237 and line 242 “(see Figure 5b, c, d, e) “ and “(Figure 7b)” should be replaced with “(see Figure 4b, c, d, e) “ and “(Figure 4b)”, the discussion refers to Figure 4 and not to Figures 5 and 7.

Thank you for pointing this out. I agree with this comment. Therefore, "I have corrected the icon. This change can be found in the revised manuscript In the Section 3.3. line 237 and line 242.

The authors should correct the title of Figure 5. Instead of CMDA replace CDMA.

Thank you for pointing this out. I agree with this comment. Therefore, I have corrected the spelling. This change can be found in the revised manuscript In line 254 on page 8.

Figure 7 should be placed within Section 3.8.

Thank you for pointing this out. I agree with this comment. Therefore, I have adjusted the figure to the correct position. This change can be found in the revised manuscript in Line 362 on page 12.

Line 360. The last chapter should be renamed in Conclusion, it is not Materials and Methods.

Thank you for pointing this out. I agree with this comment. Therefore, I have corrected it.

This change can be found in the revised manuscript in Line 362 on page 12.

Thank you again, Professor, for taking the time to provide me with your valuable feedback. I wish you happiness every day.

Reviewer 2 Report

Comments and Suggestions for Authors

I have carefully reviewed the manuscript "Preparation and Performance Study of Dual-Network Photo-Curable Conductive Silk Fibroin Composite Hydrogel." This work presents valuable insights into the development and characterization of an innovative composite hydrogel system. While the core concept builds upon existing research, the thorough characterization and detailed analysis make this study a worthwhile contribution to the field. The authors provide comprehensive characterization data and thoughtful discussion of their findings. The dual-network approach combined with photo-curable properties offers interesting possibilities for biomedical applications.

1. The introduction would benefit from incorporating recent relevant work on PEDOT:PSS-based conductive inks, particularly:

·         https://pubs.rsc.org/en/content/articlelanding/2020/tb/d0tb00627k/unauth

·         https://www.mdpi.com/2073-4360/13/3/474

·         https://onlinelibrary.wiley.com/doi/abs/10.1002/jbm.a.36314

2. The UV curing protocol requires additional detail:

- Specific wavelength used

- Duration of exposure

- Light intensity parameters

3. Experimental Design

Please explain:

- The reasoning behind the selected PEDOT:PSS concentration range (0.25-0.75 wt%)

- How these concentrations compare to similar systems in literature

4. Results Analysis

Consider expanding:

- The underlying mechanisms driving conductivity enhancement

- A more extensive comparison with performance metrics of similar materials

- Discussion of current material limitations and potential solutions

- Data on stability beyond the 30-day timepoint

5. Applications and Impact

The conclusion would be strengthened by:

- Elaborating on specific potential applications

- Discussing how the material properties align with requirements for these applications

- Identifying key challenges to be addressed in future work

Author Response

  1. The introduction would benefit from incorporating recent relevant work on PEDOT:PSS-based conductive inks, particularly:

https://pubs.rsc.org/en/content/articlelanding/2020/tb/d0tb00627k/unauth

https://www.mdpi.com/2073-4360/13/3/474

https://onlinelibrary.wiley.com/doi/abs/10.1002/jbm.a.36314

Thank you for pointing this out. I agree with this comment. Therefore, I have inserted the references into the paper. This change can be found in the revised manuscript in line 20 on page 1, in line 54 on page 2. and in line 83 on page 3.

  1. The UV curing protocol requires additional detail:

- Specific wavelength used

- Duration of exposure

- Light intensity parameters

Thank you for pointing this out. I agree with this comment. Therefore, I have added it to the text. This change can be found in the revised manuscript in line 115 on page 4.

  1. Experimental Design

Please explain:

- The reasoning behind the selected PEDOT:PSS concentration range (0.25-0.75 wt%)

- How these concentrations compare to similar systems in literature.

Thank you for pointing this out. I agree with this comment. Therefore, This is to ensure that the material is non-toxic and has good electrical conductivity. I have added it to the text.

This change can be found in the revised manuscript in line 91 on page 3.

  1. Results Analysis

Consider expanding:

- The underlying mechanisms driving conductivity enhancement

- A more extensive comparison with performance metrics of similar materials

- Discussion of current material limitations and potential solutions

- Data on stability beyond the 30-day timepoint

    However, due to the Chinese New Year, the laboratory is closed, and I am unable to conduct experiments at the moment. I hope you can understand, and I will definitely follow your suggestions in my future research.

  1. Applications and Impact

The conclusion would be strengthened by:

- Elaborating on specific potential applications

- Discussing how the material properties align with requirements for these applications

- Identifying key challenges to be addressed in future work

Thank you for pointing this out. I agree with this comment. Therefore, I have added it to the text. This change can be found in the revised manuscript in line 373 on page 12.

Thank you again, Professor, for taking the time to provide me with your valuable feedback. I wish you happiness every day.

Reviewer 3 Report

Comments and Suggestions for Authors

The authors' work is interesting, but some points need clarification. I also find the organization of the manuscript a little bit odd. Please add some references in some of your statements. All comments can be found in the attached pdf.

Author Response

1.Page 1, line 13: The keyword “Sil-MA” I think it must be changed to something like an actual word, such as silk fibroin or modified one.

Thank you for pointing this out. I agree with this comment. Therefore, I have changed the keyword to SF. This change can be found in the revised manuscript in line 12 on page 1.

  1. Page 1, lines 17-18: Please refer some examples.

Thank you for pointing this out. I agree with this comment. Therefore, I have added some examples. This change can be found in the revised manuscript in line 18 on page 1.

  1. Introduction: I am missing some examples of conductive polymers apart the one the authors are using. Moreover, what about other works related with gels with incorporated conductive materials. I think you need to show some works in order to see what is actually under research.

Thank you for pointing this out. I agree with this comment. However, my current research focuses on improving electrical conductivity and mechanical properties. After making improvements in other areas, I will certainly add them.

  1. 2.2 methodology: If a protocol from another work was used, please refer it.

Thank you for pointing this out. I agree with this comment. Therefore, I have added the references. This change can be found in the revised manuscript in line 75 on page 3.

  1. Figure 1 caption: What do the authors mean with this?

    Thank you for pointing this out. This figure is to demonstrate the gel preparation process.

  1. I think it is missing a Table with all the samples and the analogies used.

Thank you for pointing this out. I agree with this comment. Currently, it is only a preliminary study with limited data, which makes it inconvenient to present in tables. After further investigation, I will set up orthogonal experiments for deeper exploration.

  1. Page 6, line 184: The authors state that the band gap is decreased, but they do not state for which set of materials. What is the band gap for all three set of materials?

    Thank you for pointing this out. I agree with this comment. I have indicated the materials in paper. This change can be found in the revised manuscript in line 190 on page 6.

  1. 3.2: Some references are needed.

Thank you for pointing this out. I agree with this comment. Therefore, I have added the references. This change can be found in the revised manuscript in line 204 on page 6.

  1. Figure 3: The change for sample CDMA5 is quite evident, but for samples CDMA3 vs CDMA1 is not. What is the reason for no significant changes between simples CDMA1-3?

Thank you for pointing this out. At the molecular level, when molecules aggregate, interactions between them (such as van der Waals forces, hydrogen bonds, etc.) may occur, which could lead to a decrease in the total energy of the system, thereby affecting the material's electronic structure. As molecules aggregate into larger units, the interactions between these molecules may lead to a redistribution of the electron cloud, resulting in a decrease in electronic energy.

  1. Page 7, lines 213-223: Please add references

Thank you for pointing this out. I agree with this comment. Therefore, I have added the references. This change can be found in the revised manuscript in line 222 on page 7.

  1. Figure 4b: Which CDMA is this?

    Thank you for pointing this out. Here, we are discussing the secondary structures of different materials. The purple bar chart represents CDMA.

  1. Page 8, lines 255-258: Please add references.

Thank you for pointing this out. I agree with this comment. Therefore, I have added the references. This change can be found in the revised manuscript in line 267 on page 8.

  1. . Authors are referring CDMA with two different nomenclatures, 0.25-0.75 and 1 3. This is confusing. Are these the same materials? Please explain better.

Thank you for pointing this out. I agree with this comment. They are the same thing. Due to carelessness in my previous work, I hadn't corrected it, but it has now been corrected. Thank you, Professor.

  1. Page 8, lines 262-263: How did they calculate this?

Thank you for pointing this out. This was simulated and calculated using ImageJ, as mentioned in line 268 on page 8 of the text.

  1. Page 8, lines 263-267: Please add references.

Thank you for pointing this out. I agree with this comment. Therefore, I have added the references. This change can be found in the revised manuscript in line 272 on page 8.

  1. Figure 5: I think this figure should be rearranged. The authors talk only for the SEM morphology and the other parts of the figure are described in the next session. See also line 237 in page 7 and line 242 in page 8. The authors should have an order in the figures presented and being in context with what is described.

Thank you for pointing this out. I agree with this comment. Therefore, I have readjusted the position. This change can be found in the revised manuscript ion page 9.

  1. Page 9, lines 271-275: Please add references.

Thank you for pointing this out. I agree with this comment. Therefore, I have added the references. This change can be found in the revised manuscript in line 267 on page 8.

  1. Figure 5: The images e-g cannot be found inside the text. The authors need to describe everything they have in their figures.

Thank you for pointing this out. I agree with this comment. Therefore, I have provided a description for the image. This change can be found in the revised manuscript in line 291 on page 9.

  1. Page 10, lines 292-296: Please add references.

Thank you for pointing this out. I agree with this comment. Therefore, I have added the references. This change can be found in the revised manuscript in line 272 on page 8.

  1. Figure 7 is placed before the text, please put the figures in accordance to the text.

Thank you for pointing this out. I agree with this comment. Therefore, I have moved it to the relevant position. This change can be found in the revised manuscript on page 12.

  1. Page 13, line 360: Here the authors mean “Conclusions”.

Thank you for pointing this out. I agree with this comment. Therefore, I have corrected it. This change can be found in the revised manuscript in line 371 on page 12.

Thank you again, Professor, for taking the time to provide me with your valuable feedback. I wish you happiness every day.
